# Mediating effect of smoking abstinence self-efficacy on association between health literacy and nicotine dependence of smokers in Qingdao, China

Yani Wang[1], Fei Qi[1], Jie Yang[2], Haiyan Xu[3], Aimiao Tian[2], Shasha Fang[3], Gongli Liu[4], Yaoqi Zhang[4], Shanpeng Li[1]*, Kunzheng Lv[1]*

1 Qingdao Municipal Center for Disease Control and Prevention, Qingdao, Shandong, China,
2 Community Health Service Center of XueJiadao Street, Qingdao, Shandong, China, 3 Community Health Service Center of Chengyang Street, Qingdao, Shandong, China, 4 School of Public Health, Qingdao University, Qingdao, Shandong, China

☻ These authors contributed equally to this work.
* lshpeng@163.com (SL); qdjkjy@126.com (KL)

## Abstract

### Objective

Research on the mediating effect of smoking abstinence self-efficacy (SASE) on the relationship between health literacy and nicotine dependence is limited. This study examines the mechanisms through which health literacy influences addictive behavior and provides additional scientific evidence to inform smoking cessation interventions.

### Methods

A total of 500 participants were recruited from 20 communities in Qingdao between June 2023 and June 2024. Spearman correlation analysis was used to explore relationships between nicotine dependence and other factors. Linear regression analyses were conducted to examine the relationships between health literacy, SASE, and nicotine dependence. Path analysis was performed using AMOS to assess the interactions among nicotine dependence, SASE, and health literacy, with mediation effects tested using the bootstrap method.

### Results

After adjusting for potential confounders such as age, marital status, education level, and occupation, path analysis revealed a significant positive correlation between health literacy and SASE in habitual/addictive situations (SASEH; β = 0.185, p < 0.01). Nicotine dependence showed significant negative correlations with both health literacy (β = −0.289, p < 0.001) and SASEH (β = −0.513, p < 0.001). Bootstrap mediation

**Data availability statement:** Data cannot be shared publicly because the data used in our study were obtained from the Qingdao Municipal Center for Disease Control and Prevention, which is managed by the government. Government departments allow researchers to use the data for scientific research, but do not allow anyone to share original data publicly.Data are available from the science and technology department of Qingdao Municipal Center for Disease Control and Prevention(contact via E-mail: cdcjkjy@qd.shandong.cn.) for researchers who meet the criteria for access to confidential data.

**Funding:** This research is supported by the Qingdao Outstanding Health Professional Development Fund; the Medical and Health Scientific Research Project of Qingdao(2023-WJZD123)-A Study on a Comprehensive Community-based Smoking Cessation Intervention Applying Appropriate Techniques of Traditional Chinese Medicine based on Social Cognitive Theory; and the Medical and Health Technology Project of Shandong (202312050847)-A Study on a Comprehensive Community-based Smoking Cessation Intervention based on Health Belief Model.

**Competing interests:** The authors have declared that no competing interests exist.

tests confirmed that both the direct and indirect effects were statistically significant, with the mediating effect accounting for 24.7% of the total effect.

## Conclusions

SASEH partially mediates the relationship between health literacy and nicotine dependence. Increasing health literacy not only directly reduces nicotine dependence but also improves SASE, which in turn further reduces dependence.

---

## Introduction

Tobacco use is a major public health concern. Smoking can cause cardiovascular disease, respiratory illness, and several forms of cancer, particularly lung cancer [1]. It contributes to the loss of 57 million disability-adjusted life years and is among the top 10 risk factors for mortality [2].The World Health Organization's (WHO) 2019 report on the global tobacco epidemic states that tobacco use results in 8 million deaths each year worldwide [3]. China consumes more than one-third of the world's cigarettes [4]. The WHO Framework Convention on Tobacco Control and the "MPOWER" strategy emphasize the provision of smoking cessation services as an effective tobacco control approach. Consistent with this, the Healthy China 2030 Plan aims to strengthen national tobacco control measures and smoking cessation support. Despite ongoing efforts, the 2018 China Adult Tobacco Survey Report indicated that the smoking prevalence among adults aged 15 years and older was 26.59% [5]. Only 6.63% of current smokers planned to quit within the next month, and 17.96% had attempted to quit during the past year [6].

Smoking initiation is shaped by social, psychological, and physiological factors that contribute to sustained use, dependence development, and relapse during exposure to addictive substances [7]. Evidence has shown that nicotine dependence strongly affects smoking cessation success [8–10].Individuals with higher dependence levels often find cessation and long-term abstinence more difficult [8–10]. Moreover, groups with lower levels of education, income, or occupational status, as well as racial and ethnic minorities, show disproportionately high smoking rates [11–13]. These groups also face barriers to cessation due to limited access to cessation resources [14, 15]. Health literacy, defined as the ability to obtain, understand, and apply health information to make informed health decisions, has shown a negative association with nicotine dependence [16–18]. Individuals with lower health literacy generally have higher nicotine dependence and lower cessation success. Smoking abstinence self-efficacy (SASE), defined as confidence in refusing smoking in high-risk situations, also shows a negative association with nicotine dependence [19]. SASE has been widely applied in international smoking cessation research [20]. Jiang's study reported that increasing awareness of smoking-related harms can promote cessation success and enhance SASE, thereby supporting abstinence maintenance [21].

Recent research has largely examined single smoking hazard awareness rather than broader health literacy. However, health literacy is a multifaceted construct in

which health knowledge forms the foundation for additional competencies such as understanding, evaluating, and using health information to guide actions that support health. Although health knowledge is recognized as important, the present study focuses on how broader health literacy contributes to smoking cessation. Research examining the mediating effect of SASE on the association between health literacy and nicotine dependence remains limited. Therefore, this study proposes the following hypotheses: ① Lower health literacy is associated with higher nicotine dependence; ② Higher health literacy is associated with higher SASE; ③ SASE mediates the effect of health literacy on nicotine dependence. Based on these hypotheses, a mediation analysis model is constructed to examine the mediating effect of SASE, providing a theoretical foundation for developing tobacco control intervention strategies.

## Methods

### Design and participants

This study used baseline data from the "Multi-center Study on a Comprehensive Community-based Smoking Cessation Intervention," a randomized controlled trial that explored the effectiveness of different smoking cessation methods and their influencing factors. A total of 500 participants were recruited from 20 communities in Qingdao between June 2023 and December 2023. All participants were smokers who were willing to quit within one month. The inclusion criteria for participants were: (1) current smokers, whether daily or occasional; (2) intention to quit smoking within one month; (3) age 18 years or older; and (4) agreement to participate in the survey and completion of the informed consent form. Participants were excluded if they had conditions, such as mental health disorders, that could hinder their participation in the intervention program. All 500 participants completed the questionnaires. To ensure data validity and authenticity, the questionnaires were filled out anonymously, with each participant assigned a randomly generated number for anonymity. The data were accessed for research purposes between January 2024 and August 2025. The authors had no access to personally identifiable information during or after data collection.

### Ethical approval

All participants provided written informed consent. The study procedures were reviewed and approved by the Ethics Committee of the Chinese Center for Disease Control and Prevention (approval number: 202314) and registered with the Chinese Clinical Trial Registry(ChiCTR2400080614) on February 2, 2024.

### Questionnaire content and scale

**Sociodemographic questionnaire.** The sociodemographic characteristics collected included: age, gender (male/female), marital status (unmarried/married/divorced/widowed), education level (elementary school and below/junior high school/high school or vocational school/junior college/college or higher), occupation (government or public institution staff/enterprise, commercial, and service industry staff/farmer/medical staff/retiree/unemployed/other), and other relevant information.

**Fagerström test of nicotine dependence (FTND).** The nicotine dependence was assessed using the six-item scale revised by Heatherton et al [22].The scale includes the following items: (1) How soon after you wake up do you smoke your first cigarette? (scored 0–3); (2) Do you find it difficult to refrain from smoking in places where it is forbidden? (scored 0–1); (3) Which cigarette would you most hate to give up? (scored 0–1); (4) How many cigarettes do you smoke per day? (scored 0–3); (5) Do you smoke more frequently during the first hours after waking than during the rest of the day? (scored 0–1); (6) Do you smoke even when you are so ill that you spend most of the day in bed? The total score on the FTND ranges from 0 to 10, with higher scores indicating greater nicotine dependence. Scores of 0–3, 4–6, and ≥7 correspond to mild, moderate, and severe dependence, respectively. In this study, the Cronbach's alpha coefficient for the nicotine dependence test scale was 0.728.

**Smoking abstinence self-efficacy (SASE).** SASE was measured using the nine-item scale developed by Velicer et al [18]. This scale assesses participants' confidence in their ability to resist smoking in various situations. The scale consists of three dimensions: (1) positive/social SASE (SASEP), which includes scenarios like socializing with friends, drinking tea or relaxing, or spending time with a spouse or close friend who smokes; (2) negative/affective SASE (SASEN), which includes situations such as feeling anxious, angry, or depressed; and (3) habitual/addictive SASE (SASEH), which includes moments like waking up in the morning, needing a pick-me-up, or feeling the urge after not smoking for a whereas. The scale uses a five-point Likert scale, ranging from 1 ("extremely want to smoke") to 5 ("do not want to smoke at all"), with higher scores indicating higher levels of SASE. The Cronbach's alpha coefficient for the SASE scale in this study was 0.926.

**Health literacy.** The simplified health literacy scale developed by Sun et al. [23] consists of four items across three dimensions: healthcare, health promotion, and disease prevention. It uses a five-point Likert scale, ranging from 1 ("very difficult") to 4 ("very easy"). The total score on the scale ranges from 4 to 16, with higher scores indicating higher levels of health literacy. In this study, the Cronbach's alpha coefficient for the health literacy scale was 0.888.

## Statistical methods

IBM SPSS 27.0 was used to calculate descriptive statistics and correlations for the key variables. Normally distributed data are presented as mean±standard deviation, whereas data with an abnormal distribution are reported as median and quartiles [M(Q1, Q3)]. Nonparametric tests, such as the Mann-Whitney U test or Kruskal-Wallis test, were used for variables with abnormal distributions.

For the mediation effect analysis, the initial steps included assessing the data quality and evaluating the model's appropriateness. This was done by checking if the data were suitable for factor analysis using the Kaiser-Meyer-Olkin (KMO) measure and Bartlett's test of sphericity. The KMO measure is a statistical index used to determine the adequacy of the sample for factor analysis. It quantifies the proportion of partial correlation coefficients between variables, helping to evaluate whether the sample data can effectively extract common factors [24, 25]. The following KMO value ranges are considered: 0.9 and above (very suitable for factor analysis), 0.8–0.9 (suitable for factor analysis), 0.7–0.8 (acceptable for factor analysis), 0.6–0.7 (barely acceptable—factor analysis should be carefully considered), and below 0.5 (unsuitable for factor analysis) [24].

The reliability of the scales was assessed using Cronbach's alpha coefficient. Additionally, Harman's single-factor test was used to assess common method bias in the questionnaires, helping to identify any potential defects in the measurement tools, data collection methods, or analysis techniques [26]. Spearman correlation analysis was conducted to explore the relationships between nicotine dependence and other factors in the model. Path analysis was performed using AMOS, which was employed to examine the relationships between nicotine dependence, SASE, and health literacy. Model adjustments were made using the Bollen-Stine method. AMOS, a popular software for covariance-based structural equation modeling (CB-SEM), is closely integrated with IBM SPSS. It is designed to establish and verify complex models of causal relationships between variables, including path analysis and confirmatory factor analysis. AMOS can handle complex multivariable relationships and extend analysis capabilities, such as identifying abnormal impact cases through its plug-in function [27]. Finally, mediation testing was conducted using the bootstrap method, a non-parametric resampling technique that does not rely on the assumption of normality for the sampling distribution. Bootstrapping entailed repeatedly sampling from the dataset to estimate the indirect effect in each resampled dataset. This process was repeated 5,000 times to create an empirical approximation of the sampling distribution of the indirect effect, providing confidence intervals (CIs) for the indirect effect [28]. A p-value of <0.05 was considered statistically significant. Mediation is a critical concept in social science research, where a variable (M) mediates the effect of an independent variable (X) on a dependent variable (Y). The purpose of mediation analysis is to determine whether the relationship between X and Y is partly or entirely explained by the mediator variable M [29].

 

The total effect of health literacy on nicotine dependence refers to the effect of health literacy on nicotine dependence without considering the mediating variable, SASE. This total effect can be divided into two components: the direct effect, which is the effect of health literacy on nicotine dependence after controlling for SASE, and the indirect effect, also known as the mediating effect. The indirect effect refers to the influence of health literacy on nicotine dependence through the mediating variable, SASE.

## Results

### Basic demographic characteristics

This study included 500 participants, the majority of whom were male (99.8%). The age distribution was fairly balanced: 27.0% were between 18 and 44 years old, 36.2% were between 45 and 59 years old, and 36.8% were over 60 years old. Most participants were married (93.4%). In terms of education, 42.0% of participants had completed junior high school, 25.6% had attended high school or vocational school, and 15.4% had completed elementary school or less. Regarding occupation, the largest group consisted of staff members in enterprises, commercial, and service industries (45.6%), followed by retirees (20.6%) (Table 1).

### Reliability and validity of the questionnaire

A preliminary reliability and validity test was conducted on the overall questionnaire. The KMO value was 0.899, and Bartlett's test of sphericity yielded a p-value of <0.001, indicating good validity and suitability for factor analysis. Given that all research variables were measured through self-report questionnaires, the possibility of common method bias was considered. To address this, Harman's single-factor analysis was employed to test for common method bias. Four factors with eigenvalues greater than 1 were extracted, with the largest eigenvalue accounting for 36.267% of the total variance, which is below the critical threshold of 40%. Therefore, no significant issue of common method bias was found in this study [30].

### SASE of the participants

Significant differences were found in the SASEP scores based on education level (P <0.001). Participants with a college education or higher scored significantly higher (9.364 ± 1.941) compared to those with a junior college education [6(6, 9), Z = −3.823, P<0.001], high school/vocational school education [7(6, 9), Z = −3.401, P<0.001], junior high school education [7(6, 9), Z = −3.536, P<0.001], and elementary school or below [7(6, 9), Z = −4.012, P<0.001].

Significant differences were also found in the SASEN scores across occupations (P=0.013). Unemployed participants had significantly lower SASE scores [6 (6, 7.5)] compared to government/public institution staff (8.059 ± 2.045, Z = −2.282, P=0.023); enterprise, commercial, and service industry staff [7 (6, 9), Z = −2.266, P=0.023]; farmers [8 (6, 9), Z = −2.861, P=0.004]; and retirees [7 (6, 9), Z = −2.064, P=0.039] (Table 1).

### FTND of the participants

Significant differences were found in the FTND scores among smokers of different age groups (P=0.005). Participants aged ≥60 years had significantly higher FTND scores [3 (1, 6)] compared to those aged 45–59 years [3 (1, 5), Z = −2.401, P=0.016] and 18–44 years [2 (1, 4.5), Z = −3.013, P=0.003]. Significant variations in FTND scores were also observed across marital status (P=0.021). Bereaved participants had significantly higher FTND scores (5.900 ± 2.846) compared to unmarried participants [2 (0, 5), Z = −2.706, P=0.006] and married participants [3 (1, 5), Z = −2.985, P=0.003]. Significant differences were found in FTND scores across education levels (P=0.001). Participants with a college education or higher had significantly lower FTND scores [0.5 (0, 2)] compared to those with junior college education [2 (0, 4.5), Z = −2.589, P=0.018], high school/vocational education [2 (1, 5), Z = −3.267, P=0.001], junior high school education [3 (1, 5), Z = −3.631, P<0.001], and elementary school or below [3 (1, 6), Z = −3.579, P<0.001].

**Table 1. Comparison of SASE and FTND among smokers with different demographic characteristics.**

| Characteristics | Categories | N | % | Total score of SASE Mean±SD/ M(Q1, Q3) | P | Positive/ Social situations Mean±SD/ M(Q1, Q3) | P | Habit/ Addictive situations Mean±SD/ M(Q1, Q3) | P | Negative/ Affective situations Mean±SD/ M(Q1, Q3) | P | FTND Mean±SD/ M(Q1, Q3) | P |
|---|---|---|---|---|---|---|---|---|---|---|---|---|---|
| Gender[a] | Male[a] | 499 | 99.8 | 23(18,28) | 0.241 | 7(6,9) | 0.952 | 7(6,10) | 0.097 | 7(6,9) | 0.395 | 3(1,5) | 0.156 |
| | Female | 1 | 0.2 | 18 | | 7 | | 5 | | 6 | | 7 | |
| Age (year)[b] | 18-44 | 135 | 27.0 | 24(19,27) | 0.634 | 8(6,9) | 0.543 | 9(6,10) | 0.544 | 7(6,9) | 0.861 | 2(1,4.5) | 0.005 |
| | 45-59 | 181 | 36.2 | 23(18,28) | | 7(6,9) | | 9(6,10) | | 7(6,9) | | 3(1,5) | |
| | ≥60 | 184 | 36.8 | 23(18,27) | | 7(6,9) | | 9(6,9.5) | | 7(6,9) | | 3(1,6) | |
| Marriage[b] | Unmarried | 18 | 3.6 | 25.110±7.020 | 0.651 | 8.111±2.763 | 0.792 | 9.278±2.782 | 0.088 | 7.722±2.469 | 0.892 | 2(0,5) | 0.021 |
| | Married | 467 | 93.4 | 23(18,28) | | 6(7,9) | | 9(6,10) | | 7(6,9) | | 3(1,5) | |
| | Divorced | 5 | 1.0 | 23.600±5.230 | | 6(6,9) | | 8.000±2.121 | | 8.400±2.510 | | 3.600±2.881 | |
| | Bereave | 10 | 2.0 | 22.100±4.795 | | 7.100±2.234 | | 7.200±1.398 | | 9(6,9) | | 5.900±2.846 | |
| Education levels[b] | Elementary school and below | 77 | 15.4 | 22.779±5.984 | 0.034 | 7(6,9) | <0.001 | 9(6,9) | 0.095 | 7(6,9) | 0.568 | 3(1,6) | 0.001 |
| | Junior high school | 210 | 42.0 | 23(19,27) | | 7(6,9) | | 9(6,10) | | 7(6,9) | | 3(1,5) | |
| | High school/ vocational school | 128 | 25.6 | 24(18,28) | | 7(6,9) | | 9(6,10) | | 8(6,9) | | 2(1,5) | |
| | junior college | 63 | 12.6 | 22(19,27) | | 6(6,9) | | 9(6.5,10) | | 7(6,9) | | 2(0,4.5) | |
| | College and above | 22 | 4.4 | 28.318±7.345 | | 9.364±1.941 | | 10(7,12) | | 9(6,11) | | 0.5(0,2) | |
| Occupation[b] | Government/ Public Institution Staff | 17 | 3.4 | 23(21.5,27) | 0.221 | 7.941±1.983 | 0.554 | 9(8,9) | 0.617 | 8.059±2.045 | 0.013 | 2(1,3) | 0.009 |
| | Enterprise, Commercial, and Service Industry Staff | 228 | 45.6 | 22.5(18,29) | | 7(6,9) | | 9(6,10) | | 7(6,9) | | 2(1,5) | |
| | Farmer | 65 | 13.0 | 23(20,27) | | 7(6,9) | | 9(7,9) | | 8(6,9) | | 4(2,6) | |
| | Medical staff | 11 | 2.2 | 22.727±8.742 | | 7.545±2.252 | | 7.909±3.390 | | 7.273±3.438 | | 0(0,5) | |
| | Retiree | 103 | 20.6 | 23(18,27) | | 7(6,9) | | 9(6,10) | | 7(6,9) | | 3(1,5) | |
| | Unemployed | 27 | 5.4 | 21(20,23) | | 7(6,8) | | 9(8,9) | | 6(6,7.5) | | 3.556±2.026 | |
| | Others | 45 | 9.0 | 27(18,29.5) | | 8(6,9) | | 9(6,12) | | 9(6,10) | | 3(0,4) | |

[a]Mann-Whitney U test; [b]Kruskal-Wallis test.

Significant differences were also observed in FTND scores among smokers with different occupations (P = 0.009). Farmers had significantly higher FTND scores [4 (2, 6)] compared to enterprise, commercial, and service industry staff [2 (1, 5), Z = −3.087, P = 0.002] (Table 1).

**Correlation of health literacy, SASE and FTND**

The correlation analysis of the variables shown in Table 2 revealed several significant relationships. Nicotine dependence was negatively correlated with health literacy (r = −0.276, p < 0.001), the total SASE score (r = −0.463, p < 0.001), as well

**Table 2. Correlation analysis among health literacy, SASE, and nicotine dependence.**

| | health literacy | SASE | Positive/social situations | Negative/affective situations | Habituated/addictive situations | nicotine dependence |
|---|---|---|---|---|---|---|
| Health literacy | 1 | – | – | – | – | – |
| SASE | 0.144** | 1 | – | – | – | – |
| Positive/social situations | 0.145** | 0.849*** | 1 | – | – | – |
| Negative/affective situations | 0.113* | 0.896*** | 0.680*** | 1 | – | – |
| Habituated/addictive situations | 0.168*** | 0.909*** | 0.666*** | 0.736*** | 1 | – |
| Nicotine dependence | −0.276*** | −0.463*** | −0.378*** | −0.371*** | −0.486*** | 1 |

*p<0.05, **p<0.01, ***p<0.001.

as with the three subscales of SASE: positive/social situations (r= −0.378, p<0.001), negative/affective situations (r= −0.371, p<0.001), and habituated/addictive situations (r= −0.486, p<0.001). Health literacy was positively correlated with the total SASE score (r=0.144, p<0.001), positive/social SASE (r=0.145, p<0.01), negative/affective SASE (r=0.113, p<0.05), and habituated/addictive SASE (r=0.168, p<0.001).

The regression analysis results in Table 3 showed a negative correlation between health literacy and FTND ($\beta$= −0.191, p<0.001), as well as a significant negative correlation between the habituated/addictive situations subscale of SASE (SASEH) and FTND ($\beta$= −0.408, p<0.001). Together, four factors (SASEP, SASEN, SASEH, and health literacy) accounted for 23.9% of the variance in FTND among the participants ($R^2$=0.239, p<0.001). However, SASEN and SASEP did not significantly affect FTND. Therefore, they were excluded from the mediation analysis path of health literacy, SASE, and nicotine dependence.

A path analysis was conducted on the hypothesized model using structural equation modeling (SEM). Following Bollen and Stine's recommendations [31], the model was adjusted, resulting in fit indices of $\chi^2$/df=2.925, goodness-of-fit index (GFI)=0.933, adjusted GFI=0.901, Tucker–Lewis index=0.923, comparative fit index=0.942, and root mean square error of approximation (RMSEA)=0.062. All indices met the standard criteria, thereby indicating a good model fit [32, 33]. After adjusting for potential confounders such as age, marital status, education level, and occupation, the path analysis results (shown in Table 4 and Fig 1) revealed a significantly positive correlation between health literacy and SASEH ($\beta$=0.185, p<0.0 1). Nicotine dependence showed significantly negative correlations with both health literacy ($\beta$= −0.289, p<0.001)

**Table 3. Regression analysis of variables in the model.**

| The regression equation | | Index | | Significance of regression coefficient | |
|---|---|---|---|---|---|
| The results of variable | Predictor variable | R2 | F | β | t |
| Total score of SASE | Health literacy | 0.034 | 18.640*** | 0.190 | 4.317*** |
| Positive/social situations | Health literacy | 0.027 | 15.071*** | 0.171 | 3.882*** |
| Negative/affective situations | Health literacy | 0.023 | 12.562*** | 0.157 | 3.544*** |
| Habituated/addictive situations | Health literacy | 0.032 | 17.594*** | 0.185 | 4.195*** |
| FTND | Positive/social situations | 0.239 | 40.268*** | −0.037 | −0.644 |
| | Negative/affective situations | | | 0.016 | 0.240 |
| | Habituated/addictive situations | | | −0.408 | −6.412*** |
| | Health literacy | | | −0.191 | −4.800*** |

***p<0.001.

**Table 4. Mediating effect of SASE in habituated/addictive situations on the relationship between health literacy and nicotine dependence.**

| Effect | β | 95%CI | p | Proportion of Effect |
|---|---|---|---|---|
| Total effect | −0.384 | (−0.528,-0.242) | <0.001 | |
| Indirect effect | −0.095 | (−0.171,-0.037) | 0.001 | 24.7% |
| Direct effect | −0.289 | (−0.428,-0.141) | <0.001 | 75.3% |

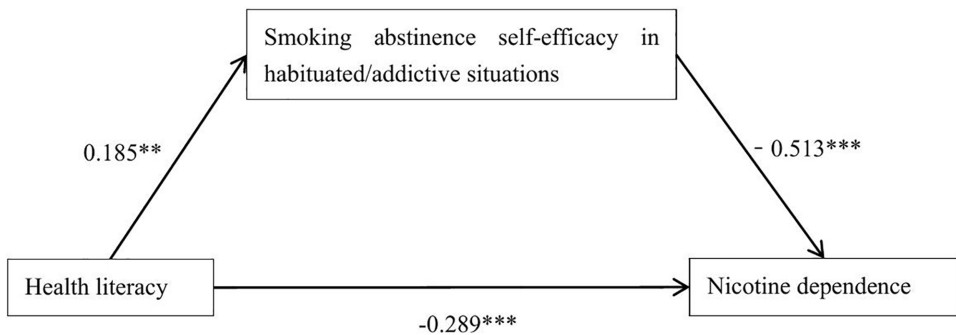

**Fig 1. Mediating effect pathway between health literacy and nicotine dependence via SASE in habituated/addictive situations.**
**p<0.01,***p<0.001.

and SASEH (β = −0.513, p<0.001). The bootstrap mediation test (Table 4) indicated that both the direct and indirect effects were statistically significant, with the mediating effect accounting for 24.7% of the total effect.

## Discussion

Research on the mediating effect of SASE in the relationship between health literacy and nicotine dependence is still limited. This study, therefore, sought to explore the associations between health literacy, SASE, and nicotine dependence among smokers who were motivated to quit. Moreover, it aimed to identify the factors influencing SASE and nicotine dependence.

The results showed that participants with a college education or higher had significantly higher SASE scores compared to those with lower education levels. This finding aligns with previous studies [34, 35]. Educated individuals typically have a broader knowledge base and better information-processing skills, which may help them better understand the dangers of smoking, including the associated health risks. This understanding, in turn, likely fosters stronger self-efficacy in resisting tobacco use. Population data from the National Health Interview Survey in the U.S. suggests that higher education levels are strongly associated with successful smoking cessation [36]. Furthermore, individuals with higher education levels tend to have larger social support networks, such as friends, family, and colleagues. These networks can provide positive reinforcement and encouragement, which further enhances their SASE in social situations. The study also found that unemployed participants had lower SASEN compared to those in other occupations. Unemployment often brings significant economic strain, a crisis of social identity, and uncertainty about the future. These factors can lead to emotional distress, such as anxiety and depression, which may prompt smokers to turn to cigarettes as a means of coping. This, in turn, lowers their SASE. Previous research has suggested that addressing smokers' negative beliefs about quitting can improve smoking cessation success [37]. Therefore, improving smokers' ability to manage negative emotions and encouraging alternative coping behaviors, other than smoking, are essential strategies to enhance their self-efficacy.

This study also identified several demographic factors, including age, marital status, education level, and occupation, that are closely associated with nicotine dependence. Bereaved participants showed higher levels of nicotine dependence than their married and unmarried counterparts. Bereavement often triggers emotional challenges, such as depression and anxiety, and smoking may be used as a way to manage these negative feelings. Numerous studies have shown that individuals with depression are more likely to smoke and develop nicotine dependence [38–41]. In contrast, strong family support has been shown to help reduce the risk of depression and smoking-related issues [42, 43]. The results also confirmed that higher education levels were associated with lower nicotine dependence, consistent with prior research. Smokers with higher education tend to have greater awareness of the risks associated with tobacco use and more proactive behaviors toward quitting [44]. On the other hand, smokers with lower education levels may lack awareness about tobacco hazards, leading to misconceptions and a reduced focus on health, which can contribute to heavier smoking and stronger dependence [7]. Moreover, farmers had higher levels of nicotine dependence compared to participants in other occupations. This may be due to the physically demanding nature of their work, with smoking serving as a means of relieving fatigue and stress. Moreover, smoking is often a social activity in rural areas of China, where it is commonly used as a tool for building relationships and networking [45]. Therefore, how to improve the health literacy level and enhance their SASE has become a key focus in increasing the smoking cessation rate for high-risk populations(e.g., elderly smokers, those with low education, farmers, and bereaved individuals).

The findings also highlighted that individuals with lower health literacy had higher levels of nicotine dependence, which is consistent with previous research [16]. Studies have shown that improving self-efficacy can increase smoking cessation rates and reduce nicotine dependence. SASE is a key predictor of smoking cessation success and relapse prevention [46, 47]. Thus, tobacco control programs and health communication campaigns should focus on delivering information about the dangers of smoking whereas also providing strategies to improve smokers' self-efficacy.

Moreover, the study found that individuals with low health literacy were more likely to engage in smoking behaviors and exhibit higher nicotine dependence. In contrast, high health literacy was associated with increased SASEH (self-efficacy in habituated/addictive situations), which in turn contributed to reduced nicotine dependence. Improving smokers' health literacy not only directly reduces their nicotine dependence but also enhances their SASE, creating an indirect reduction in dependence. Smokers with higher health literacy generally have a better understanding of tobacco-related health risks, which strengthens their motivation to quit smoking and improves their confidence in their ability to succeed. Therefore, efforts should focus on increasing smokers' health literacy and providing comprehensive smoking cessation strategies to reduce nicotine dependence. By highlighting the role of health literacy and smoking abstinence self-efficacy (SASE) in reducing nicotine dependence, it guides public health campaigns to prioritize health education and self-efficacy building, improving population health outcomes.

In future, our research can develop tailored programs for high-risk subgroups (e.g., unemployed individuals with low SASEN, elderly smokers) to address their unique barriers to cessation. Future longitudinal studies can verify causal relationships between the variables, while intervention studies can test whether improving health literacy and SASE reduces nicotine dependence over time.

## Limitations and future directions

This study has several limitations. First, it relied on cross-sectional data, which, whereas useful for generating hypotheses, does not allow for conclusions about causality. Therefore, the findings should be interpreted with caution. Longitudinal studies are necessary to better understand the temporal relationships between health literacy and nicotine dependence. Second, since the data were self-reported, there is a risk of selection bias and recall bias. Lastly, although the analysis adjusted for key demographic and socioeconomic variables, there may still be residual confounding from unmeasured or unknown factors that could influence the results. Therefore, future research should consider conducting longitudinal studies to further investigate the relationships between health literacy, SASE, nicotine dependence, and smoking cessation outcomes.

## Conclusion

The findings of this study revealed that participants with a college education or higher exhibited the highest levels of SASE, particularly in SASEP, compared to those with lower education levels. Unemployed participants, on the other hand, had lower scores on the SASEN subscale. Additionally, age, marital status, education level, and occupation were found to be closely related to nicotine dependence among smokers. Participants aged 60 and above, those who were bereaved, and those with lower education levels were found to have higher levels of nicotine dependence. The study also showed that SASE partially mediates the relationship between health literacy and nicotine dependence. Improving the health literacy of smokers can not only reduce their nicotine dependence directly but also enhance their SASE, leading to an indirect reduction in dependence.

## Acknowledgments

Thanks to the China CDC for all the support of this study. Gratefully thank all the investigators and participants in the survey for their valuable contributions to this study. Thanks to the staff of Chengyang District Community Health Service Center in Qingdao who assisted in the data collection.

## Author contributions

**Data curation:** Aimiao Tian.

**Formal analysis:** Shasha Fang.

**Investigation:** Haiyan Xu, Shanpeng Li.

**Methodology:** Yani Wang, Fei Qi, Shasha Fang, Gongli Liu.

**Project administration:** Kunzheng Lv.

**Resources:** Jie Yang, Haiyan Xu, Shasha Fang, Shanpeng Li, Kunzheng Lv.

**Software:** Yaoqi Zhang.

**Validation:** Gongli Liu.

**Visualization:** Gongli Liu.

**Writing – original draft:** Yani Wang, Yaoqi Zhang.

**Writing – review & editing:** Yani Wang.

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
