## [Decision Letter · Decision Letter 0]

1 Oct 2025

Dear Dr. Wang,

Thank you for submitting your manuscript to PLOS ONE. After careful consideration, we feel that it has merit but does not fully meet PLOS ONE’s publication criteria as it currently stands. Therefore, we invite you to submit a revised version of the manuscript that addresses the points raised during the review process.

**ACADEMIC EDITOR:**
**Please find the reviewers excellent comments attached. Please revise the manuscript and resubmit for further consideration for publication**

We look forward to receiving your revised manuscript.

Kind regards,

Souparno Mitra, M.D.

Academic Editor

PLOS ONE

Journal Requirements:

[This research is supported by the Qingdao Outstanding Health Professional Development Fund; the Medical and Health Scientific Research Project of Qingdao(2023-WJZD123)-A Study on a Comprehensive Community-based Smoking Cessation Intervention Applying Appropriate Techniques of Traditional Chinese Medicine based on Social Cognitive Theory; and the Medical and Health Technology Project of Shandong (202312050847)-A Study on a Comprehensive Community-based Smoking Cessation Intervention based on Health Belief Model.].

3. Thank you for stating the following in your manuscript:

[This research is supported by the Qingdao Outstanding Health Professional Development Fund; the Medical and Health Scientific Research Project of Qingdao(2023-WJZD123)-A Study on a Comprehensive Community-based Smoking Cessation Intervention Applying Appropriate Techniques of Traditional Chinese Medicine based on Social Cognitive Theory; and the Medical and Health Technology Project of Shandong (202312050847)-A Study on a Comprehensive Community-based Smoking Cessation Intervention based on Health Belief Model.]

[This research is supported by the Qingdao Outstanding Health Professional Development Fund; the Medical and Health Scientific Research Project of Qingdao(2023-WJZD123)-A Study on a Comprehensive Community-based Smoking Cessation Intervention Applying Appropriate Techniques of Traditional Chinese Medicine based on Social Cognitive Theory; and the Medical and Health Technology Project of Shandong (202312050847)-A Study on a Comprehensive Community-based Smoking Cessation Intervention based on Health Belief Model.]

5. Please amend your list of authors on the manuscript to ensure that each author is linked to an affiliation. Authors’ affiliations should reflect the institution where the work was done (if authors moved subsequently, you can also list the new affiliation stating “current affiliation:….” as necessary).

Reviewers' comments:

Reviewer's Responses to Questions

**Comments to the Author**

1. Is the manuscript technically sound, and do the data support the conclusions?

Reviewer #1: Partly

Reviewer #2: Partly

2. Has the statistical analysis been performed appropriately and rigorously?

Reviewer #1: Yes

Reviewer #2: Yes

3. Have the authors made all data underlying the findings in their manuscript fully available?

Reviewer #1: Yes

Reviewer #2: Yes

4. Is the manuscript presented in an intelligible fashion and written in standard English?

Reviewer #1: Yes

Reviewer #2: Yes

Reviewer #1: This study examines the impact mechanism of health literacy on addictive behavior, and attempts to provide some scientific evidence for smoking cessation intervention. The authors warn that the information relied on self reported cross-sectional data and that although these results may be useful for generating hypotheses, they do not imply causality and should be interpreted cautiously. Also residual confounding may result from unmeasured or unknown factors and may affect the result.

With these limitations in mind the paper uses the appropriate statistical analysis which appears suitable for this undertaking. The paper is generally well written statistically. However, in several instances the investigators provide insufficient statistical detail for explaining the terminology and results. It would help the reader and interpretation of the results to fill in these gaps. For example:

1. What is Bootstrap mediation?

2. Specifically what motivated the KMO measure for this data? Define method bias for the reader. Define the Kaiser– Meyer–Olkin (KMO) measure. For this study the KMO measure is 0.889. Explain the acceptable range for the KMO and interpretation.

3. Please comment on the sample size of 500 and rationale for that number and its suitability of size in light of this pathway approach which appears to have an acceptable number of predictors in this context.

4. Define AMOS for the reader.

5. Getting back to the limitations, How generalizable are these results given that the targeted sample was only for college educated only?

6. Despite the statistical significance of the R-squares in Table 3 , please explain the clinical impact of these factors.

7. In Table 3 , how do the regression coefficients change if some of the factors were entered stepwise in or out of the model and how would Figure 1 path correlations be affected?

Reviewer #2: I am grateful for the opportunity to review this work. The following represent the most important opportunities I see for strengthening this manuscript:

1- Abstract the structure should still be in line with an empirical research format. The abstract should be revisited.

2- Poor citation format; including initials in citations is common in the manuscript.

The authors made poor attempt at espousing the gap in knowledge; that the previous authors did not come up with 'conceptual framework'. The authors should rather look at the results and see how that can be extended or flaws noticed in those studies.

3- Common method variance. Is there a possibility of common method bias, as self-reported questionnaires measured all study variables? This bias may affect the accuracy and robustness of the results.

4- In what ways does this research benefit both society as whole and existing fields of study? Use how this research can in future.

5- Structure of content (not of headings), is not easy to follow. For example, the first point should lead to the second. Some sentences are too long; and the grammar and English language needs attention.

I hope this helps. In the current state, I cannot recommend the paper for publication, until these revisions are completed satisfactorily.

**Do you want your identity to be public for this peer review?** For information about this choice, including consent withdrawal, please see our Privacy Policy

Reviewer #1: No

Reviewer #2: **Yes:** Mai Helmy

---

## [Author Response · Author response to Decision Letter 1]

17 Nov 2025

Dear editors,

Thank you very much for your letter and the reviewers’ constructive comments concerning our manuscript entitled “Mediating effect of smoking abstinence self-efficacy on association between health literacy and nicotine dependence of smokers in Qingdao, China” (Manuscript ID: PONE-D-25-44023). We have carefully considered all the comments and made revisions to the manuscript accordingly. We believe that the revisions have significantly improved the quality of our paper.

We have provided a point-by-point response to all the reviewers’ comments in the following pages. All changes in the revised manuscript have been highlighted using the “Track Changes” function in Microsoft Word.

In addition, we appreciate the reviewer's suggestion regarding data sharing. The data used in our study were obtained from the Qingdao Municipal Center for Disease Control and Prevention, which is managed by the government. Government departments allow researchers to use the data for scientific research, but do not allow anyone to share original data publicly. Interested researchers may contacted the science and technology department of Qingdao Municipal Center for Disease Control and Prevention via the following contact information:E-mail: cdcjkjy@qd.shandong.cn.

We would like to express our sincere gratitude to you and the reviewers for the thoughtful feedback and the opportunity to improve our work. We hope that the revised manuscript is now acceptable for publication in PLOS ONE.

Sincerely,

Yani Wang

Qingdao Municipal Center for Disease Control and Prevention

+86 17854299979

Responses to Reviewers’ Comments

Reviewer #1:

Comments from the Reviewer

1.What is Bootstrap mediation?

Authors’ Response We thank the reviewer for this helpful suggestion. We have now added a detailed description of Bootstrap mediation.

Changes in Manuscript Page 4, Lines 165-170.

2.Specifically what motivated the KMO measure for this data? Define method bias for the reader. Define the Kaiser–Meyer–Olkin (KMO) measure. For this study the KMO measure is 0.889. Explain the acceptable range for the KMO and interpretation.

Authors’ Response We thank the reviewer for this valuable suggestion. We have now added a detailed description of the KMO measure and method bias.

Changes in Manuscript We have added the definition and the acceptable range for the KMO in page 4, Lines 147-153. We have added the definition of method bias in page 4, 154-157.

3.Please comment on the sample size of 500 and rationale for that number and its suitability of size in light of this pathway approach which appears to have an acceptable number of predictors in this context.

Authors’ Response Sample size determination was guided by SEM requirements, with recommendations ranging from 250 to 500 for complex models [1], with a 5-10:1 case-to-parameter ratio [2], and allowing a 20% nonresponse rate [3]. This study includes 19 projects, so at least 228 research objects are required, and 500 research objects are sufficient.

[1] HOOGLAND JJ, BOOMSMA A. Robustness studies in covariance structure modeling. Sociol Methods Res. 1998;26(3):329–367. doi: 10.1177/0049124198026003003.

[2] BENTLER PM, CHOU CP. Practical issues in structural modeling. Sociol Methods Res. 1987;16(1):78–117. doi: 10.1177/0049124187016001004.

[3] Sakpal TV. Sample size estimation in clinical trial. Perspect Clin Res. 2010 Apr;1(2):67-9. PMID: 21829786; PMCID: PMC3148614.

4.Define AMOS for the reader.

Authors’ Response We have added the definition of AMOS.

Changes in Manuscript Page 4, 160-165.

5.Getting back to the limitations, How generalizable are these results given that the targeted sample was only for college educated only?

Authors’ Response We thank the reviewer for raising this important point. This study covers groups with different educational levels, such as 15.4% with Elementary school education or below, 42.0% with junior high school education, 25.6% with high school/vocational school education, 12.6% with junior college education, and 4.4% with college or above. This result has good generalizability.

6.Despite the statistical significance of the R-squares in Table 3 , please explain the clinical impact of these factors.

Authors’ Response We thank the reviewer for this valuable suggestion. We have now added the clinical impact of these factors.

Changes in Manuscript Page 7, Lines 240-241.

7. In Table 3 , how do the regression coefficients change if some of the factors were entered stepwise in or out of the model and how would Figure 1 path correlations be affected?

Authors’ Response When the stepwise regression method is used to include or eliminate some factors, the regression coefficient may change slightly and gradually, and may show a gradual decreasing trend. These changes are due to the interaction between variables to recalibrate the model, but do not affect the core findings.

Stepwise regression requires stepwise testing of the significance of paths A and B. when the mediating effect is weak (e.g. A or B is close to the critical value), the testing power is lower than that of bootstrap method, and it is prone to "false negative" conclusions.

Reviewer #2:

1.Abstract the structure should still be in line with an empirical research format. The abstract should be revisited.

Authors’ Response We thank the reviewer for this comment. The abstract have been revisited with an empirical research format.

Changes in Manuscript Page 1, 15-41.

2.Poor citation format; including initials in citations is common in the manuscript.

The authors made poor attempt at espousing the gap in knowledge; that the previous authors did not come up with 'conceptual framework'. The authors should rather look at the results and see how that can be extended or flaws noticed in those studies.

Authors’ Response We have modified the citation format in the part of “References”.

We thank the editor for this critical comment. We agree that our original justification for the knowledge gap was insufficient. As suggested, we have now reframed the introduction to base the research gap on a critical analysis of the existing empirical evidence.

Recent research has largely examined single smoking hazard awareness rather than broader health literacy. However, health literacy is a multifaceted construct in which health knowledge forms the foundation for additional competencies such as understanding, evaluating, and using health information to guide actions that support health. Although health knowledge is recognized as important, the present study focuses on how broader health literacy contributes to smoking cessation.

Changes in Manuscript Page 2, 71-76.

3.Common method variance. Is there a possibility of common method bias, as self-reported questionnaires measured all study variables? This bias may affect the accuracy and robustness of the results.

Authors’ Response When all research variables are measured by self-report questionnaire, the possibility of common method bias will increase This deviation will indeed affect the accuracy and reliability of the research results[7]. In this study, as stated in the “Results-Questionnaire test of reliability and validity” section, Harman's single factor analysis method was used to test the common method bias. The maximum eigenvalue factor accounted for 36.267% of the total variance�page 5, Lines 196�, which was 40% lower than the critical threshold. Therefore, there is no significant common method bias in this study.

4.In what ways does this research benefit both society as whole and existing fields of study? Use how this research can in future.

Authors’ Response

It provides empirical support for tobacco control policies and smoking cessation programs, helping optimize interventions targeting high-risk groups (e.g., elderly smokers, those with low education, farmers, and bereaved individuals).(Page 9, 317-320.)

By highlighting the role of health literacy and smoking abstinence self-efficacy (SASE) in reducing nicotine dependence, it guides public health campaigns to prioritize health education and self-efficacy building, improving population health outcomes.�Page 10, 337-339.

In future, our research can develop tailored programs for high-risk subgroups (e.g., unemployed individuals with low SASEN, elderly smokers) to address their unique barriers to cessation. Future longitudinal studies can verify causal relationships between the variables, while intervention studies can test whether improving health literacy and SASE reduces nicotine dependence over time.(Page 10, 340-344.)

5.Structure of content (not of headings), is not easy to follow. For example, the first point should lead to the second. Some sentences are too long; and the grammar and English language needs attention.

Authors’ Response We sincerely thank the reviewer for this valuable feedback. We agree that the flow of the manuscript could be significantly improved. In response, we have undertaken a comprehensive revision of the text to enhance its clarity and readability. Specifically, we have:

①Restructured the logical flow between ideas, particularly in the sections mentioned, to ensure that arguments progress more naturally and coherently. The connection between the first and second points has now been made explicit to guide the reader smoothly through our line of reasoning.

②Thoroughly revised the language throughout the manuscript. This involved breaking down overly long and complex sentences into shorter, more focused ones, and carefully correcting grammatical errors.

③Sought the assistance of a professional language editing service to polish the English and ensure it meets the high standards of the journal. A certificate of editing is available upon request.

We believe these extensive revisions have greatly improved the manuscript's clarity, flow, and overall quality, and we are grateful for the suggestion.

---

## [Decision Letter · Decision Letter 1]

14 Jan 2026

Mediating effect of smoking abstinence self-efficacy on association between health literacy and nicotine dependence of smokers

PONE-D-25-44023R1

Dear Dr. Wang,

We’re pleased to inform you that your manuscript has been judged scientifically suitable for publication and will be formally accepted for publication once it meets all outstanding technical requirements.

Kind regards,

Souparno Mitra, M.D.

Academic Editor

PLOS One

Additional Editor Comments (optional):

Reviewers' comments:

Reviewer's Responses to Questions

**Comments to the Author**

Reviewer #1: All comments have been addressed

Reviewer #2: All comments have been addressed

2. Is the manuscript technically sound, and do the data support the conclusions?

Reviewer #1: (No Response)

Reviewer #2: Yes

3. Has the statistical analysis been performed appropriately and rigorously?

Reviewer #1: (No Response)

Reviewer #2: Yes

4. Have the authors made all data underlying the findings in their manuscript fully available?

Reviewer #1: (No Response)

Reviewer #2: Yes

5. Is the manuscript presented in an intelligible fashion and written in standard English?

Reviewer #1: (No Response)

Reviewer #2: Yes

Reviewer #1: (No Response)

Reviewer #2: The authors have clearly attempted to address prior reviewer concerns, regarding all of the issues.

**Do you want your identity to be public for this peer review?** For information about this choice, including consent withdrawal, please see our Privacy Policy

Reviewer #1: No

Reviewer #2: **Yes:** Mai Helmy/ Sultan Qaboos University

---

## [Editor Report · Acceptance letter]

PONE-D-25-44023R1

PLOS One

Dear Dr. Wang,

I'm pleased to inform you that your manuscript has been deemed suitable for publication in PLOS One. Congratulations! Your manuscript is now being handed over to our production team.

Kind regards,

on behalf of

Dr. Souparno Mitra

Academic Editor

PLOS One